# MARINA-P: Superior Performance in Non-smooth Federated Optimization with Adaptive Stepsizes

## Abstract

Non-smooth communication-efficient federated optimization remains largely unexplored theoretically, despite its importance in machine learning applications. We consider a setup focusing on optimizing downlink communication by improving state-of-the-art schemes like EF21-P (Gruntkowska et al., 2023) and MARINA-P (Gruntkowska et al., 2024) in the non-smooth convex setting. Our key contributions include extending the non-smooth convex theory of EF21-P from single-node to distributed settings and generalizing MARINA-P to non-smooth convex optimization. For both algorithms, we prove optimal $\mathcal{O}\left(1/\sqrt{T}\right)$ convergence rates under standard assumptions and establish matching communication complexity bounds with classical subgradient methods. We provide theoretical guarantees under constant, decreasing, and adaptive (Polyak-type) stepsizes. Our experiments demonstrate MARINA-P's superior performance with correlated compressors in both smooth non-convex and non-smooth convex settings. This work presents the first theoretical analysis of distributed non-smooth optimization with server-to-worker compression, including comprehensive analysis for various stepsize schemes.

## 1 Introduction

In recent years, the machine learning community has witnessed a paradigm shift toward larger models and datasets, spurring major performance gains but also posing new hardware, algorithmic, and software challenges (LeCun et al., 2015; Bottou et al., 2018; Kaplan et al., 2020; Deng et al., 2009).

**The Rise of Big Data and Distributed Systems.** The sheer volume of data needed for cutting-edge models has driven the adoption of distributed systems (Dean et al., 2012; Khirirat et al., 2018; Lin et al., 2018), since single-machine setups can no longer handle the storage and computational demands. This approach is particularly relevant in supervised learning (Hastie et al., 2009; Shalev-Shwartz & Ben-David, 2014; Vapnik, 2013), often formulated as:

$$\min_{x \in \mathbb{R}^d} \left\{ f(x) := \frac{1}{n} \sum_{i=1}^{n} f_i(x) \right\}, \tag{1}$$

where $n$ denotes the number of clients, $x \in \mathbb{R}^d$ is the model's parameter vector, and $f_i(x)$ is the local loss on client $i$. Throughout, we assume each $f_i$ is convex (possibly non-smooth).

Federated learning (FL) (McMahan et al., 2016; Konečný et al., 2016b;a; McMahan et al., 2017) extends the distributed paradigm to heterogeneous clients with decentralized data, seeking to avoid central data aggregation and preserve privacy. In FL, devices connect to a central server that orchestrates training (Konečný et al., 2016b; Kairouz et al., 2021): each device locally updates parameters using its data, then sends these updates to the server. The server aggregates them, performs global calculations, and broadcasts new parameters back to devices. This process continues until convergence or acceptable performance is reached.

**Communication Challenges in Large-scale Model Training.** Although distributing data alleviates storage and compute constraints, it introduces substantial communication overhead. Modern gradient-based methods (Bottou, 2012; Kingma & Ba, 2014; Demidovich et al., 2023; Duchi et al., 2011; Robbins & Monro, 1951) require iterative updates for all $d$ parameters, making frequent transmission of high-dimensional gradients expensive. Two broad approaches reduce this burden: (i) performing

multiple local gradient steps before communicating, as in LocalSGD (Stich, 2020; Khaled et al., 2020; Woodworth et al., 2020; Yi et al., 2024; Sadiev et al., 2022; Richtárik et al., 2024), and (ii) compressing gradients via lossy transformations (Khirirat et al., 2018; Alistarh et al., 2018b; Mishchenko et al., 2020; 2019; Li et al., 2020; Li & Richtárik, 2021; Richtárik et al., 2021; Fatkhullin et al., 2021; Richtárik et al., 2022; Seide et al., 2014; Alistarh et al., 2017; Panferov et al., 2024). Moreover, studies of 4G LTE and 5G networks (Huang et al., 2012; Narayanan et al., 2021) show that upload/download speeds are often comparable, emphasizing that both server-to-worker and worker-to-server communication must be optimized.

**Prevalence of Non-smooth Objectives in Machine Learning Applications.** Despite notable advances in distributed optimization, theoretical work has primarily targeted smooth objectives, leaving non-smooth problems less explored in federated contexts. Non-smoothness arises in various ML scenarios: ReLU activations (Glorot et al., 2011; Nair & Hinton, 2010), L1 regularization for sparsity (Tibshirani, 1996; Zou & Hastie, 2005), hinge loss (Cortes, 1995), total variation (Rudin et al., 1992; Chambolle, 2004), quantile regression (Koenker & Bassett Jr, 1978), max-pooling (Scherer et al., 2010), submodular minimization (Bach, 2013), Huber loss (Huber, 1964), and graph-based learning (Hallac et al., 2015).

**Adaptive Stepsizes are Widely Used in Practice.** Because constants like L-Lipschitz continuity or smoothness parameters are difficult to determine in deep learning, practitioners rely on adaptive learning rates. Popular methods include AdaGrad (Duchi et al., 2011), RMSProp, Adam (Kingma & Ba, 2014), and AMSGrad (Reddi et al., 2018), all of which adjust per-parameter stepsizes based on observed gradients.

## 1.1 NOTATION AND ASSUMPTIONS

We denote the set $\{1,2,\cdots,n\}$ by $[n]$. For vectors, $\|\cdot\|_2$ represents the Euclidean norm, while for matrices, it denotes the spectral norm. The inner product of vectors $u$ and $v$ is denoted by $\langle u, v \rangle$. We use $\mathcal{O}(\cdot)$ to hide absolute constants. We denote $R_0 := \|x^0 - x^*\|_2$.

Our analysis relies on the following standard assumptions:

**Assumption 1.** *The function $f$ has at least one minimizer, denoted by $x^*$.*

**Assumption 2.** *The functions $f_i$ are convex for all $i \in [n]$.*

In the distributed setting, assuming convexity for individual functions $f_i$ is sufficient, as it implies convexity for $f$ itself.

**Assumption 3** (Lipschitz continuity of $f_i$). *Functions $f_i$ are $L_{0,i}$-Lipschitz continuous for all $i \in [n]$. That is, for all $i \in [n]$, there exists $L_{0,i} > 0$ such that $|f_i(x) - f_i(y)| \leq L_{0,i} \|x - y\|_2$, $\quad \forall x, y \in \mathbb{R}^d$.*

If each $f_i$ is Lipschitz continuous, then by Jensen's inequality, $f$ is $L_0$-Lipschitz with $L_0 := \frac{1}{n} \sum_{i=1}^{n} L_{0,i}$ (Nesterov, 2013).

Both convexity and Lipschitz continuity of $f$ are standard assumptions in non-smooth optimization (Vorontsova et al., 2021; Nesterov, 2013; Bubeck, 2015; Beck, 2017; Duchi, 2018; Lan, 2020; Drusvyatskiy, 2020). Moreover, $L_0$ and $L_{0,i}$-Lipschitz continuity imply uniformly bounded subgradients (Beck, 2017), a property that will be useful in our proofs:

$$\|\partial f(x)\|_2 \leq L_0 \quad \forall x \in \mathbb{R}^d, \tag{2}$$

$$\|\partial f_i(x)\|_2 \leq L_{0,i} \quad \forall x \in \mathbb{R}^d \text{ and } \forall i \in [n]. \tag{3}$$

We define $\widetilde{L}_0 := \sqrt{\frac{1}{n} \sum_{i=1}^{n} L_{0,i}^2}$ and $\overline{L}_0 := \frac{1}{n} \sum_{i=1}^{n} L_{0,i}$. By the arithmetic-quadratic mean inequality, we have $\overline{L}_0 \leq \widetilde{L}_0$.

Following classical optimization literature (Nemirovski et al., 2009; Beck, 2017; Duchi, 2018; Lan, 2020; Drusvyatskiy, 2020), for non-smooth convex objectives, we aim to find an $\varepsilon$-suboptimal solution: a random vector $\hat{x} \in \mathbb{R}^d$ satisfying $\mathbb{E}[f(\hat{x}) - f(x^*)] \leq \varepsilon$, where $\mathbb{E}[\cdot]$ denotes the expectation over algorithmic randomness.

To assess the efficiency of distributed subgradient-based algorithms, we primarily use two metrics:

*1. Communication complexity* (alternatively, communication cost): The expected total number of floats per worker required to communicate to reach an $\varepsilon$-suboptimal solution. In this paper, we focus on server-to-worker communication compression.

*2. Iteration complexity*: The number of communication rounds needed to achieve an $\varepsilon$-suboptimal solution.

## 1.2 Related work

**Subgradient Methods in Non-smooth Convex Optimization.** Subgradient methods, pioneered in the 1960s (Shor et al., 1985; Polyak, 1987), remain central to non-smooth convex optimization. Classic theory establishes $\mathcal{O}\left(1/\sqrt{T}\right)$ rates for general convex objectives (Nesterov, 2013; Vorontsova et al., 2021; Bubeck, 2015; Beck, 2017; Duchi, 2018; Lan, 2020; Drusvyatskiy, 2020) and $\mathcal{O}\left(1/T\right)$ for strongly convex problems (Beck, 2017; Drusvyatskiy, 2020). For unknown $T$, decreasing stepsizes of order $\mathcal{O}\left(1/\sqrt{t}\right)$ or $\mathcal{O}\left(1/t\right)$ add a logarithmic factor, yielding $\mathcal{O}\left(\log T/\sqrt{T}\right)$ (Nesterov, 2013) and $\mathcal{O}\left(\log T/T\right)$ (Hazan et al., 2007; Hazan & Kale, 2014). Nevertheless, recent works (Zhu et al., 2024; Lacoste-Julien et al., 2012; Rakhlin et al., 2011) have removed these factors, attaining optimal rates in convex and strongly convex settings. In the stochastic regime, mirror-descent methods also achieve $\mathcal{O}\left(1/\sqrt{T}\right)$ (Nemirovski et al., 2009). Beyond averaged-iterate convergence, tighter last-iterate analyses (Jain et al., 2019; Zamani & Glineur, 2023) provide stronger guarantees. Subgradient methods remain crucial for large-scale machine learning tasks, including support vector machines and structured prediction (Shalev-Shwartz et al., 2007; Ratliff et al., 2007).

**Communication Compression.** Before discussing more advanced optimization methods, let us consider the simplest baseline: the standard subgradient method (SM) [1], which iteratively performs updates [2]

$$x^{t+1} = x^t - \frac{\gamma_t}{n} \sum_{i=1}^n g_i^t, \tag{4}$$

where $g_i^t = \partial f_i(x^t)$ is a subgradient of $f_i$ at $x^t$. In the distributed setting, the method can be implemented as follows: each worker calculates $g_i^t$ and sends it to the server, where the subgradients are aggregated. The server takes the step and broadcasts $x^{t+1}$ back to the workers. With stepsize $\gamma_t := R_0/L_0\sqrt{T}$, where $R_0 := \left\|x^0 - x^*\right\|_2$ and $T$ is the total number of iterations, SM finds an $\varepsilon$-approximate solution after $\mathcal{O}\left(L_0^2 R_0^2/\varepsilon^2\right)$ steps (Nesterov, 2013; Drusvyatskiy, 2020). Since at each step the workers and the server send $\Theta(d)$ coordinates/floats, the worker-to-server and server-to-worker communication costs are $\mathcal{O}\left(dL_0^2 R_0^2/\varepsilon^2\right)$. To formally quantify communication costs, we introduce the following definition.

**Definition 1.** *The worker-to-server (w2s, uplink) and server-to-worker (s2w, downlink) communication complexities of a method are the expected number of coordinates/floats that a worker sends to the server and that the server sends to a worker, respectively, to find an $\varepsilon$–solution.*

Communication compression techniques, such as sparsification (Wang et al., 2018; Mishchenko et al., 2020; Alistarh et al., 2018b; Wangni et al., 2018; Konečný & Richtárik, 2018) and quantization (Alistarh et al., 2017; Wen et al., 2017; Zhang et al., 2016; Horváth et al., 2022; Wu et al., 2018; Mishchenko et al., 2019), are known to be immensely powerful for reducing the communication overhead of gradient-type methods. Existing literature primarily considers two main classes of compression operators: *unbiased* and *biased (contractive)* compressors.

**Definition 2.** *(Unbiased compressor).* A stochastic mapping $\mathcal{Q} : \mathbb{R}^d \to \mathbb{R}^d$ is called an unbiased compressor/compression operator if there exists $\omega \geq 0$ such that for any $x \in \mathbb{R}^d$:

$$\mathbb{E}[\mathcal{Q}(x)] = x, \quad \mathbb{E}\left[\left\|\mathcal{Q}(x) - x\right\|_2^2\right] \leq \omega \left\|x\right\|_2^2. \tag{5}$$

This definition encompasses a wide range of well-known compression techniques, including RandK sparsification (Stich et al., 2018), random dithering (Roberts, 1962; Goodall, 1951), and natural

---

[1]In this paper, we use the non-normalized form (4) of the subgradient method studied in (Vorontsova et al., 2021; Bubeck, 2015; Beck, 2017; Duchi, 2018; Lan, 2020; Drusvyatskiy, 2020; Nemirovski et al., 2009). Earlier works (Shor et al., 1985; Polyak, 1987) typically employed SM in the form $x^{t+1} = x^t - \frac{\gamma_t}{\|\partial f(x^t)\|} \partial f(x^t)$, which includes an additional normalization term $\left\|\partial f(x^t)\right\|$.

[2]For constrained optimization problems, the subgradient method typically operates through projections onto a convex set $\mathcal{X}$ (see (Bubeck, 2015; Lacoste-Julien et al., 2012; Beck, 2017; Duchi, 2018)). However, when optimizing over an unbounded domain, i.e., $\mathcal{X} = \mathbb{R}^d$, projections are not needed.

compression (Horváth et al., 2022). Notable examples of methods employing compressor (5) are QSGD (Alistarh et al., 2017), DCGD (Khirirat et al., 2018), MARINA (Gorbunov et al., 2021), DIANA (Mishchenko et al., 2019), VR-DIANA (Horváth et al., 2019), DASHA (Tyurin & Richtárik, 2023), FedCOMGATE (Haddadpour et al., 2021), FedPAQ (Reisizadeh et al., 2020), FedSTEPH (Das et al., 2020), FedCOM (Haddadpour et al., 2021), ADIANA (Li et al., 2020), NEOLITHIC (Huang et al., 2022a), ACGD (Li et al., 2020), and CANITA (Li & Richtárik, 2021). However, Definition 2 does not cover another important class of practically more favorable compressors, called *contractive*, which are usually biased.

**Definition 3.** *(Contractive compressor). A stochastic mapping $\mathcal{C} : \mathbb{R}^d \to \mathbb{R}^d$ is called a contractive compressor/compression operator if there exists $\alpha \in (0,1]$ such that for any $x \in \mathbb{R}^d$:*

$$\mathbb{E}\left[\|\mathcal{C}(x) - x\|_2^2\right] \leq (1 - \alpha) \|x\|_2^2. \tag{6}$$

We denote the families of compressors satisfying Definitions 2 and 3 by $\mathbb{U}(\omega)$ and $\mathbb{B}(\alpha)$, respectively.[3]

Inequality (6) is satisfied by many compressors, including Top$K$ (Ström, 2015; Dryden et al., 2016; Aji & Heafield, 2017; Alistarh et al., 2018b), quantization (Alistarh et al., 2017; Horváth et al., 2022), low-rank approximations (Vogels et al., 2019; 2020; Safaryan et al., 2021), and count-sketches (Ivkin et al., 2019; Rothchild et al., 2020). For broader surveys, see (Beznosikov et al., 2023; Demidovich et al., 2023; Safaryan et al., 2022). However, naive distributed SGD with biased compression (e.g., Top$K$) can diverge (Beznosikov et al., 2023). Error Feedback (EF14), introduced by Seide et al. (2014), emerged as a key technique to avert such divergence. Early theory of EF14 was confined to single-node settings (Stich et al., 2018; Alistarh et al., 2018a; Stich & Karimireddy, 2019), then expanded to distributed setting (Cordonnier, 2018; Beznosikov et al., 2023; Koloskova et al., 2020). Richtárik et al. (2021) reformulated EF14 into EF21, achieving optimal $\mathcal{O}(1/T)$ convergence for smooth non-convex problems, surpassing the previous $\mathcal{O}(1/T^{2/3})$ rate (Koloskova et al., 2020).

The EF21 framework led to multiple variants (Richtárik et al., 2022; Fatkhullin et al., 2021), including bidirectional (s2w and w2s) biased compression. Gruntkowska et al. (2023) introduced EF21-P, combining biased s2w and unbiased w2s to improve complexity in smooth strongly convex settings. Later, Gruntkowska et al. (2024) proposed MARINA-P for smooth non-convex problems, delivering sharper bounds than both EF21 and EF21-P. Concurrently, Anonymous (2024) provided the first non-smooth convergence guarantees for EF21-P, albeit restricted to single-node scenarios.

In order to express communication complexities, we will further need the following quantities.

**Definition 4** (Expected density)**.** *For the given compression operators $\mathcal{Q}(x)$ and $\mathcal{C}(x)$, we define the expected density as $\zeta_{\mathcal{Q}} = \sup_{x \in \mathbb{R}^d} \mathbb{E}\left[\|\mathcal{Q}(x)\|_0\right]$ and $\zeta_{\mathcal{C}} = \sup_{x \in \mathbb{R}^d} \mathbb{E}\left[\|\mathcal{C}(x)\|_0\right]$, where $\|y\|_0$ is the number of non-zero components of $y \in \mathbb{R}^d$.*

Notice that the expected density is well-defined for any compression operator since $\|\mathcal{Q}(x)\|_0 \leq d$ and $\|\mathcal{C}(x)\|_0 \leq d$.

The landscape of communication-efficient federated methods for non-smooth optimization is largely unexplored, with most research targeting smooth objectives or single-node settings. Below, we highlight open challenges and gaps.

Numerous works study s2w compression (Zheng et al., 2019; Gruntkowska et al., 2023; Fatkhullin et al., 2021; Philippenko & Dieuleveut, 2021; Liu et al., 2020; Gorbunov et al., 2020; Safaryan et al., 2022; Huang et al., 2022b; Horváth et al., 2022; Tang et al., 2019; Tyurin & Richtarik, 2023; Gruntkowska et al., 2024), yet almost all focus on smooth objectives. To our knowledge, only Karimireddy et al. (2019) and Anonymous (2024) offer non-smooth convex guarantees with s2w compression, and both are limited to single-node settings with minimal relevance to federated learning.

Distributed subgradient methods are well-studied, but either lack compression (Nedic & Ozdaglar, 2009; Kiwiel & Lindberg, 2001; Hishinuma & Iiduka, 2015; Zheng et al., 2022) or focus on specific operators like quantization (Xia et al., 2023; Doan et al., 2020; 2018; Xia et al., 2022; Emiola & Enyioha, 2022), covering only the w2s direction. No comprehensive treatments address s2w compression in non-smooth distributed optimization.

---

[3]Notably, it can easily be verified (see Lemma 8 in (Richtárik et al., 2021)) that if $\mathcal{Q} \in \mathbb{U}(\omega)$, then $(\omega + 1)^{-1}\mathcal{Q} \in \mathbb{B}\left((\omega + 1)^{-1}\right)$, indicating that the family of biased compressors is wider.

| Method | Non-smooth | Distributed | Compressed communications | Compression type | Adaptive stepsizes |
|---|:---:|:---:|:---:|:---:|:---:|
| EF14 (Karimireddy et al., 2019) | ✓ | ✗ | ✓ | w2s | ✗ |
| EF21-P (Anonymous, 2024) | ✓ | ✗ | ✓ | s2w | ✓ |
| MARINA-P (Gruntkowska et al., 2024) | ✗ | ✓ | ✓ | s2w | ✗ |
| SM with Polyak Stepsize (Hazan & Kakade, 2019) | ✓ | ✗ | ✗ | - | ✓ |
| SM with Quantization (Xia et al., 2023) | ✓ | ✓ | ✓ | w2s | ✗ |
| EF21-P [OURS] | ✓ | ✓ | ✓ | s2w | ✓ |
| MARINA-P [OURS] | ✓ | ✓ | ✓ | s2w | ✓ |

Table 1: Summary of optimization methods employing worker-to-server (w2s) or server-to-worker (s2w) compression schemes.

Recent works on adaptive stepsizes (Khaled et al., 2023; Defazio et al., 2023; 2024; Mishchenko & Defazio; Defazio & Mishchenko, 2023) primarily handle single-node problems. Polyak stepsizes (Polyak, 1987; Hazan & Kakade, 2019) remain popular, but current studies (Loizou et al., 2021; Oikonomou & Loizou, 2024; Jiang & Stich, 2024) often assume smoothness or lack distributed analysis. Even existing non-smooth convex results (Hazan & Kakade, 2019; Schaipp et al., 2023) remain restricted to single-node contexts.

In summary, the intersection of non-smooth optimization, communication efficiency, and federated learning remains underexplored. Our work addresses this gap by providing the first comprehensive study of distributed non-smooth optimization with s2w compression and adaptive stepsizes, while maintaining optimal convergence rates.

## 2 CONTRIBUTIONS

We now summarize our main contributions: • **Extension of EF21-P to distributed non-smooth settings.** We generalize EF21-P (Anonymous, 2024) from single-node to distributed architectures, proving optimal $\mathcal{O}(1/\sqrt{T})$ rates for both Polyak and constant stepsizes, and a suboptimal $\mathcal{O}(\log T/\sqrt{T})$ rate for decreasing stepsizes. Our communication complexity matches classical distributed subgradient methods, addressing a longstanding gap in Error Feedback theory for non-smooth problems.

• **Introduction of MARINA-P for non-smooth objectives.** Building on Gruntkowska et al. (2024), we extend MARINA-P from smooth non-convex to non-smooth convex settings, establishing optimal $\mathcal{O}(1/\sqrt{T})$ rates for constant and Polyak stepsizes, along with a suboptimal $\mathcal{O}(\log T/\sqrt{T})$ rate under decreasing steps.

• **Superior performance of MARINA-P with correlated compressors.** Empirical results show that MARINA-P, when paired with correlated compressors, surpasses EF21-P in non-smooth settings. This extends the known benefits of correlated compressors, previously shown for smooth non-convex objectives, to non-smooth convex federated tasks.

• **Support for diverse stepsize schedules.** We provide theoretical guarantees for both algorithms under constant, decreasing, and Polyak stepsizes, bridging the gap between foundational theory and practical deep learning use cases.

To our knowledge, these are the first theoretical results for distributed non-smooth optimization incorporating s2w compression and adaptive stepsizes, while achieving provably optimal convergence rates.

---

**Algorithm 1** EF21-P (distributed version)

---

1: **Input:** initial points $w^0 = x^0 \in \mathbb{R}^d$, stepsize $\gamma_0 > 0$
2: **for** $t = 0, 1, 2, \ldots, T$ **do**
3:      **for** $i = 1, \ldots, n$ **on Workers in parallel do**
4:         Receive compressed difference $\Delta^t$ from server
5:         Compute local subgradient $g_i^t = \partial f_i(w^t)$ and send it to server
6:      **end for**
7:      **On Server:**
8:      Receive $g_i^t$ from workers
9:      Choose stepsize $\gamma_t$ (can be set according to (9), (10), or (11))
10:     $x^{t+1} = x^t - \gamma_t \frac{1}{n} \sum_{i=1}^n g_i^t$
11:     Compute $\Delta^{t+1} = \mathcal{C}(x^{t+1} - w^t)$ and broadcast it to workers
12:     $w^{t+1} = w^t + \Delta^{t+1}$
13:     **for** $i = 1, \ldots, n$ **on Workers in parallel do**
14:        $w^{t+1} = w^t + \Delta^{t+1}$
15:     **end for**
16: **end for**
17: **Output:** $x^T$

---

## 3   EF21-P

We now present the first major contribution of our paper: a distributed version of EF21-P for the non-smooth setting.

Let us first recap the standard single-node EF21-P algorithm, which aims to solve (1) via the iterative process:

$$
\begin{aligned}
x^{t+1} &= x^t - \gamma_t \nabla f(w^t) \\
w^{t+1} &= w^t + \mathcal{C}^t \left( x^{t+1} - w^t \right),
\end{aligned}
\tag{7}
$$

where $\gamma_t > 0$ is a stepsize, $x^0 \in \mathbb{R}^d$ is the initial iterate, $w^0 = x^0 \in \mathbb{R}^d$ is the initial iterate shift, and $\mathcal{C}^t$ is an instantiation of a randomized contractive compressor $\mathcal{C}$ sampled at time $t$. This method was proposed as a primal[4] counterpart to the standard EF21. It has proven particularly useful in bidirectional settings where primal compression is performed on the server side, allowing for the decoupling of primal and dual compression parameter constants. For more details, we refer the reader to the original paper (Gruntkowska et al., 2023). Anonymous (2024) first extended EF21-P to the non-smooth setting. Their key modification was replacing the "smooth" update step with a "non-smooth" one: $x^{t+1} = x^t - \gamma_t \partial f(w^t)$.

They proved an optimal rates of $\mathcal{O}\left(1/\sqrt{T}\right)$ for Polyak and constant stepsizes, and a suboptimal rate of $\mathcal{O}\left(\log T/\sqrt{T}\right)$ for decreasing stepsizes, but only for the single-node regime. In Algorithm 1, we extend these results to the distributed setting, allowing for parallel computation of subgradients $\partial f(w^t)$.

At each iteration of distributed EF21-P, the workers calculate $\partial f_i(w^t)$ and transmit it to the server. The server then averages the subgradients and updates the global model $x^t$. Subsequently, it computes the compressed difference $\Delta^{t+1} = \mathcal{C}_i^t(x^{t+1} - w^t)$ and broadcasts the same vector $\Delta^{t+1}$ to all workers. Both the server and workers then use $\Delta^{t+1}$ to update their internal states $w^t$. Note that this procedure ensures that the states $w^t$ remain synchronized between workers and the server.

We now present the convergence result for our distributed EF21-P algorithm.

**Theorem 1.** *Let Assumptions 1, 2 and 3 hold. Define a Lyapunov function $V^t := \|x^t - x^*\|_2^2 + \frac{1}{\lambda_* \theta} \|w^t - x^t\|_2^2$, where $\lambda_* := \frac{\sqrt{1-\alpha}}{1-\sqrt{1-\alpha}}$ and $\theta := 1 - \sqrt{1-\alpha}$. Define also a constant $B_* := 1 + 2\frac{\sqrt{1-\alpha}}{1-\sqrt{1-\alpha}}$. Let $\{w^t\}$ be the sequence produced by EF21-P (Algorithm 1). Define $\overline{w}^T := \frac{1}{T} \sum_{t=0}^{T-1} w^t$ and $\widehat{w}^T := \frac{1}{\sum_{t=0}^{T-1} \gamma_t} \sum_{t=0}^{T-1} \gamma_t w^t$.*

---

[4]Since it operates in the primal space of model parameters

**1. (Constant stepsize).** *If $\gamma_t := \gamma > 0$, then*

$$\mathbb{E}\left[f(\overline{w}^T) - f(x^*)\right] \leq \frac{V^0}{2\gamma T} + \frac{B_* L_0^2 \gamma}{2}. \tag{8}$$

*If, moreover, optimal $\gamma$ is chosen i.e.*

$$\gamma := \frac{1}{\sqrt{T}} \sqrt{\frac{V^0}{B_* L_0^2}}, \tag{9}$$

*then $\mathbb{E}\left[f(\overline{w}^T) - f(x^*)\right] \leq \frac{\sqrt{B_* L_0^2 V^0}}{\sqrt{T}}$.*

**2. Polyak stepsize.** *If $\gamma_t$ is chosen as*

$$\gamma_t \quad := \quad \frac{f(w^t) - f(x^*)}{B_* \|\partial f(w^t)\|_2^2}, \tag{10}$$

*then $\mathbb{E}\left[f(\overline{w}^T) - f(x^*)\right] \leq \frac{\sqrt{B_* L_0^2 V^0}}{\sqrt{T}}$.*

**3. (Decreasing stepsize).** *If $\gamma_t$ is chosen as*

$$\gamma_t := \frac{\gamma_0}{\sqrt{t+1}}, \tag{11}$$

*then $\mathbb{E}\left[f(\widehat{w}^T) - f(x^*)\right] \leq \frac{V^0 + 2\gamma_0^2 B_* L_0^2 \log(T+1)}{\gamma_0 \sqrt{T}}$.*

*If, moreover, optimal $\gamma_0$ is chosen i.e.*

$$\gamma_0 := \sqrt{\frac{V_0}{2B_* L_0^2 \log(T+1)}}, \tag{12}$$

*then $\mathbb{E}\left[f(\widehat{w}^T) - f(x^*)\right] \leq 2\sqrt{2B_* L_0^2 V_0}\sqrt{\frac{\log(T+1)}{T}}$.*

Let us analyze the obtained results. The constant $B_* := 1 + 2\frac{\sqrt{1-\alpha}}{1-\sqrt{1-\alpha}} \leq \frac{4}{\alpha} - 1$ is a decreasing function in $\alpha$, which aligns with intuition since larger values of $\alpha$ correspond to less aggressive compression regimes. For both constant (9) and Polyak (10) stepsizes, we achieve the optimal rate of $\mathcal{O}\left(1/\sqrt{T}\right)$ known for uncompressed subgradient methods (Nesterov, 2013; Arjevani & Shamir, 2015). However, achieving this rate requires either knowing the total number of iterations $T$ in advance (for constant stepsize) or knowing the optimal value $f(x^*)$ (for Polyak stepsize), which may be impractical in many applications. When neither $T$ nor $f(x^*)$ is known, one can employ the decreasing stepsize strategy (11). This approach leads to a suboptimal convergence rate of $\mathcal{O}\left(\log T/\sqrt{T}\right)$ – a well-known limitation of subgradient methods (Nesterov, 2013; Anonymous, 2024).

For both constant and Polyak stepsizes, the following corollary provides explicit complexity bounds, characterizing both the number of iterations and the total communication cost needed to obtain an $\varepsilon$-approximate solution.

**Corollary 1.** *Let the conditions of the Theorem 1 are met. If $\gamma_t$ is set according to (9) or (10) (constant or Polyak stepsizes) then* EF21-P *(Algorithm 1) requires $T = \mathcal{O}\left(\frac{L_0^2 R_0^2}{\alpha \varepsilon^2}\right)$ iterations/communication rounds in order to achieve $\mathbb{E}\left[f(\overline{w}^T) - f(x^*)\right] \leq \varepsilon$, and the expected total communication cost per worker is $\mathcal{O}\left(d + \zeta_{\mathcal{C}} T\right)$.*

Let us analyze the implications of Corollary 1. In the uncompressed case ($\alpha = 1$), our algorithm achieves the optimal rate of standard Subgradient Methods (SM) (Nesterov, 2013) for first-order non-smooth optimization. With Top$K$ compression ($\zeta_{\mathcal{C}} = K$), the communication complexity becomes $\mathcal{O}\left(dL_0^2 R_0^2/\varepsilon^2\right)$, matching the worst-case complexity of distributed SM. This indicates that EF21-P with Top$K$ compression cannot improve upon SM's complexity regardless of the compression parameter $\alpha$ – a fundamental limitation in communication-compressed non-smooth optimization. Our findings align with Balkanski & Singer (2018), who demonstrated that parallelization provides no worst-case benefits for non-smooth optimization.

From a practical perspective, EF21-P's main limitation stems from broadcasting identical compressed differences $\Delta_t$ to all workers, potentially leading to poor approximations of $x^{t+1}$ by $w^t + \Delta_t$. The MARINA-P algorithm (Gruntkowska et al., 2024), originally developed for smooth non-convex problems, addresses this limitation. In the following section, we extend MARINA-P to the non-smooth setting.

---

**Algorithm 2** MARINA-P

---

1: **Input:** initial point $x^0 \in \mathbb{R}^d$, initial model shifts $w_i^0 = x^0$ for all $i \in [n]$, stepsize $\gamma_0 > 0$, probability $0 < p \leq 1$, compressors $\mathcal{Q}_i^t \in \mathbb{U}(\omega)$ for all $i \in [n]$
2: **for** $t = 0, 1, \ldots, T$ **do**
3:      **for** $i = 1, \ldots, n$ **on Workers in parallel do**
4:          Compute local subgradient $g_i^t = \partial f_i(w_i^t)$ and send it to server
5:      **end for**
6:      **On Server:**
7:      Receive $g_i^t$ from workers
8:      Choose stepsize $\gamma_t$ (can be set according to (13), (14), or (15))
9:      $x^{t+1} = x^t - \gamma_t \frac{1}{n} \sum_{i=1}^n g_i^t$
10:      Sample $c^t \sim \text{Bernoulli}(p)$
11:      **if** $c^t = 0$ **then**
12:          Send $\mathcal{Q}_i^t(x^{t+1} - x^t)$ to worker $i$ for $i \in [n]$
13:      **else**
14:          Send $x^{t+1}$ to all workers
15:      **end if**
16:      **for** $i = 1, \ldots, n$ **on Workers in parallel do**
17:          $w_i^{t+1} = \begin{cases} x^{t+1} & \text{if } c^t = 1 \\ w_i^t + \mathcal{Q}_i^t(x^{t+1} - x^t) & \text{if } c^t = 0 \end{cases}$
18:      **end for**
19: **end for**
20: **Output:** $x^T$

---

## 4 MARINA-P

Building upon the foundations of the standard MARINA algorithm (Gorbunov et al., 2021; Szlendak et al., 2022), Gruntkowska et al. (2024) introduced MARINA-P, a primal counterpart designed to operate in the model parameter space. This section presents an extension of MARINA-P to the non-smooth convex setting. At each iteration, workers compute subgradients $\partial f_i(w_i^t)$ and transmit them to the server. The server aggregates these subgradients and updates the global model $x^t$. With probability $p$ (typically small), the server sends the uncompressed updated model $x^{t+1}$ to all workers. Otherwise, each worker $i$ receives a compressed vector $\mathcal{Q}_i^t(x^{t+1} - x^t)$. Workers then update their local models $w_i^{t+1}$ based on the received information. A key feature of MARINA-P is that the compressed vectors $\mathcal{Q}_1^t(x^{t+1} - x^t), \ldots, \mathcal{Q}_n^t(x^{t+1} - x^t)$ can differ across workers. This distinction is crucial for the algorithm's practical superiority, as it allows for potentially better approximations of $x^{t+1}$ compared to methods like EF21-P, especially when the compressors $\mathcal{Q}_1, \ldots, \mathcal{Q}_n$ are correlated.

We now present the main convergence results for MARINA-P in the non-smooth convex setting.

**Theorem 2.** *Let Assumptions 1, 2 and 3 hold. Define a Lyapunov function* $V^t := \|x^t - x^*\|_2^2 + \frac{1}{\lambda_* p} \frac{1}{n} \sum_{i=1}^n \|w_i^t - x^t\|_2^2$, *where* $\lambda_* := \frac{\overline{L}_0}{\widetilde{L}_0} \sqrt{\frac{(1-p)\omega}{p}}$. *Define also a constant* $\widetilde{B}_* := \overline{L}_0^2 + 2\overline{L}_0 \widetilde{L}_0 \sqrt{\frac{(1-p)\omega}{p}}$. *Let* $\{w_i^t\}$ *be the sequence produced by* MARINA-P *(Algorithm 2). Define* $\overline{w}_i^T := \frac{1}{T} \sum_{t=0}^{T-1} w_i^t$ *and* $\widehat{w}_i^T := \frac{1}{\sum_{t=0}^{T-1} \gamma_t} \sum_{t=0}^{T-1} \gamma_t w_i^t$ *for all* $i \in [n]$.

***1. (Constant stepsize).*** *If* $\gamma_t := \gamma > 0$, *then* $\mathbb{E}\left[\frac{1}{n} \sum_{i=1}^n f_i(\overline{w}_i^T) - f(x^*)\right] \leq \frac{V^0}{2\gamma T} + \frac{\widetilde{B}_* \gamma}{2}$.

*If, moreover, optimal* $\gamma$ *is chosen i.e.*

$$\gamma := \frac{1}{\sqrt{T}} \sqrt{\frac{V^0}{\widetilde{B}_*}}, \tag{13}$$

*then* $\mathbb{E}\left[\frac{1}{n} \sum_{i=1}^n f_i(\overline{w}_i^T) - f(x^*)\right] \leq \frac{\sqrt{\widetilde{B}_* V^0}}{\sqrt{T}}$.

***2. Polyak stepsize.*** *If* $\gamma_t$ *is chosen as*

$$\gamma_t := \frac{\frac{1}{n} \sum_{i=1}^n f_i(w_i^t) - f(x^*)}{\left\|\frac{1}{n} \sum_{i=1}^n \partial f_i(w_i^t)\right\|_2^2 \left(1 + 2\frac{\sqrt{\frac{1}{n} \sum_{i=1}^n \|\partial f_i(w_i^t)\|_2^2}}{\left\|\frac{1}{n} \sum_{i=1}^n \partial f_i(w_i^t)\right\|_2} \sqrt{\frac{(1-p)\omega}{p}}\right)}, \tag{14}$$

---

then $\mathbb{E}\left[\frac{1}{n}\sum_{i=1}^{n} f_i(\overline{w}_i^T) - f(x^*)\right] \leq \frac{\sqrt{\widetilde{B}_* V^0}}{\sqrt{T}}$.

**3. (Decreasing stepsize).** *If $\gamma_t$ is chosen as*

$$\gamma_t := \frac{\gamma_0}{\sqrt{t+1}}, \tag{15}$$

*then* $\mathbb{E}\left[\frac{1}{n}\sum_{i=1}^{n} f_i(\widehat{w}_i^T) - f(x^*)\right] \leq \frac{V^0 + 2\gamma_0^2 \widetilde{B}_* \log(T+1)}{\gamma_0 \sqrt{T}}$.

*If, moreover, optimal $\gamma_0$ is chosen i.e.*

$$\gamma_0 := \sqrt{\frac{V_0}{2\widetilde{B}_* \log(T+1)}}, \tag{16}$$

*then* $\mathbb{E}\left[\frac{1}{n}\sum_{i=1}^{n} f_i(\widehat{w}_i^T) - f(x^*)\right] \leq 2\sqrt{2\widetilde{B}_* V_0}\sqrt{\frac{\log(T+1)}{T}}$.

**Remark 1.** *For both* EF21-P *and* MARINA-P*, the Polyak stepsize can be efficiently implemented in the distributed setting without additional per-iteration communication overhead. This is because the subgradient values $\partial f_i(w^t)$ (for* EF21-P*) and $\partial f_i(w_i^t)$ (for* MARINA-P*) are already computed by the clients and transmitted to the server as part of the algorithm's regular operations.*

Let us analyze these results. The constant $\widetilde{B}_* := \overline{L}_0^2 + 2\overline{L}_0\widetilde{L}_0\sqrt{\frac{(1-p)\omega}{p}}$ depends on both compression parameters $\omega$ and $p$. Smaller values of $\omega$ correspond to less aggressive compression, while larger values of $p$ indicate more frequent uncompressed communication – both cases lead to smaller $\widetilde{B}_*$ and consequently faster convergence. For both constant (13) and Polyak (14) stepsizes, we obtain the optimal rate of $\mathcal{O}\left(1/\sqrt{T}\right)$ (Nesterov, 2013; Arjevani & Shamir, 2015). As with EF21-P, achieving this rate requires either knowing the total iterations $T$ (for constant stepsize) or the optimal value $f(x^*)$ (for Polyak stepsize) in advance. When such knowledge is unavailable, the decreasing stepsize strategy offers a practical alternative, though it results in a suboptimal $\mathcal{O}\left(\log T/\sqrt{T}\right)$ convergence rate – a characteristic limitation of subgradient methods (Nesterov, 2013). It is worth noting that implementing the Polyak stepsize only requires an estimate of $f(x^*)$, rather than knowledge of the Lipschitz constant $L_0$. This characteristic is common among Polyak stepsizes (Loizou et al., 2021).

For the constant and Polyak stepsize regimes, the following corollary establishes complexity bounds and characterizes the communication costs required to achieve an $\varepsilon$-approximate solution.

**Corollary 2.** *Let the conditions of the Theorem 2 are met and $p = \zeta_{\mathcal{Q}}/d$. If $\gamma_t$ is set according to (13) or (14) (constant or Polyak stepsizes) then* MARINA-P *(Algorithm 2) requires $T = \mathcal{O}\left(\frac{R_0^2}{\varepsilon^2}\left(\overline{L}_0^2 + \overline{L}_0\widetilde{L}_0\sqrt{\omega\left(\frac{d}{\zeta_{\mathcal{Q}}} - 1\right)}\right)\right)$ iterations/communication rounds in order to achieve $\mathbb{E}\left[\frac{1}{n}\sum_{i=1}^{n} f_i(\overline{w}_i^T) - f(x^*)\right] \leq \varepsilon$, and the expected total communication cost per worker is $\mathcal{O}\left(d + \zeta_{\mathcal{Q}}T\right)$.*

This corollary reveals several important properties. With Rand$K$ compression ($\zeta_{\mathcal{Q}} = K, \omega = d/K - 1$) (Beznosikov et al., 2023), MARINA-P achieves communication complexity $\mathcal{O}\left(d\widetilde{L}_0^2 R_0^2/\varepsilon^2\right)$. Under the condition $\widetilde{L}_0^2 = \mathcal{O}\left(L_0\right)$, this matches the optimal per-worker complexity of standard SM, up to constant factors (Nesterov, 2013). A notable feature of our complexity result is its independence from the number of workers $n$ in the non-smooth setting – a known phenomenon in subgradient methods (Arjevani & Shamir, 2015; Balkanski & Singer, 2018). This contrasts with MARINA-P's behavior in smooth non-convex settings (Gruntkowska et al., 2024), where complexity scales as $\mathcal{O}(1/n)$. The absence of theoretical bounds predicting such scaling behavior in non-smooth distributed settings presents an interesting direction for future research.

MARINA-P's primary advantage over EF21-P lies in its ability to employ worker-specific compression operators $\mathcal{Q}_i$, enabling more accurate approximations of the global model, particularly when using correlated compressors. The following section examines various constructions of $\mathcal{Q}_i$ that leverage this flexibility to enhance practical performance.

## 5 IMPACT STATEMENT

This paper presents work whose goal is to advance the field of Machine Learning. There are many potential societal consequences of our work, none which we feel must be specifically highlighted here.

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
