# OpenReview forum: "MARINA-P: Superior Performance in Nonsmooth Federated Optimization with Adaptive Stepsizes"
_ICLR.cc/2026/Conference — ICLR 2026 Conference Withdrawn Submission_

### Official Review · Reviewer_vakc · 2025-10-29

**Soundness:** 3
**Presentation:** 3
**Contribution:** 2
**Rating:** 4
**Confidence:** 4

**Summary:**

This paper studies the distributed optimization problem under server-to-worker (s2w) compression with error feedback. This paper focuses on Lipschitz continuous objectives. The authors analyze two algorithms: EF21-P, which employs server-side compression, and MARINA-P, a loopless variant that uses server-side unbiased quantizers. Theoretical analyses are provided for three types of stepsize: constant, decreasing, and Polyak stepsize.

**Strengths:**

1. The theorems are written clearly and backed by reasonable proofs and analysis.
2. The synthetic experiment is well-defined for illustrations of the theoretical results.

**Weaknesses:**

1. The paper only considers s2w compression, while w2s communication is typically considered much more expensive than s2w broadcast [1,2]. It seems to make more sense to consider w2s or bidirectional compression. The current setting is a bit less interesting.

2. The main contribution is mostly an extension of prior works from the single-node to distributed settings. However, since the paper assumes the bounded gradient assumption, the typical challenge of the data heterogeneity issue (when considering smooth objectives) disappears. Therefore, it seems extending the results to the multiple client setting is natural, and the overall contribution is a bit limited to me.

3. The scale of the experiment is a bit too small to show the superiority of the algorithms compared with other algorithms.

[1] EF21: A new, simpler, theoretically better, and practically faster error feedback, Neurips 2021.

[2] EF21 with Bells & Whistles: Six Algorithmic Extensions of Modern Error Feedback. JMLR 2025.

**Questions:**

1. Could the authors clarify why MARINA-P is analyzed under unbiased compression rather than the more general contractive compression setting?


P.S. The reference: Error feedback for smooth and nonsmooth convex optimization with constant, decreasing, and polyak stepsizes, seems broken.

---

### Official Review · Reviewer_oeeX · 2025-10-31

**Soundness:** 2
**Presentation:** 2
**Contribution:** 2
**Rating:** 2
**Confidence:** 3

**Summary:**

This paper extends EF21-P and MARINA-P to non-smooth convex federated optimization with server-to-worker compression and adaptive (Polyak) stepsizes. It establishes optimal convergence rates of $O(1/\sqrt{T})$ for constant and Polyak stepsizes, and $O(\log T/\sqrt{T})$ for decreasing ones, matching the best-known subgradient bounds. Experiments show that MARINA-P performs competitively and maintains strong communication efficiency.

**Strengths:**

1. Provides the first theoretical analysis of communication-efficient federated optimization for non-smooth convex problems with s2w compression.

2. Achieves optimal convergence rates, extending EF21-P and MARINA-P frameworks with adaptive stepsizes.

3. Experiments, though limited, confirm robust performance and communication savings.

**Weaknesses:**

1. While the theoretical extension to non-smooth settings is novel, the core algorithmic structure (EF21-P and MARINA-P) is largely inherited from prior works. The contribution lies primarily in analysis, rather than introducing fundamentally new mechanisms.

2. The experiments are not extensive enough to fully substantiate claims of “superior performance.” For example, no ablation is presented on key parameters such as compression aggressiveness $(\alpha, \omega)$ or probability $p$, and the tasks tested are relatively simple.

3. The paper emphasizes theory but does not address how these results translate to real-world federated systems, where non-smooth objectives (e.g., L1 regularization or hinge losses) are used in large-scale training. Practical scalability, stability under high heterogeneity, and wall-clock communication savings are not discussed.

4. The introduction justifies the non-smooth setting generally, but it could better articulate why optimizing s2w compression in non-smooth regimes matters in practice, especially since upload and download speeds are often asymmetric in FL deployments.

5. The paper’s experiments primarily replicate setups from earlier MARINA or EF21 works, and the benefit of correlated compressors, though noted, is not deeply analyzed.

**Questions:**

1. Could the authors include experiments varying the compression ratio or probability $p$ to empirically demonstrate how communication complexity scales?

2. The theoretical results assume convex objectives; can the authors comment on the extension to non-convex but Lipschitz objectives (e.g., ReLU networks)?

3. Are there practical datasets (e.g., text or healthcare) where the proposed methods show meaningful speedups? A more applied experiment could strengthen the motivation.

4. How does MARINA-P behave under client heterogeneity or unbalanced data distributions? Some robustness evaluation would be valuable.

5. Could the authors clarify whether adaptive Polyak stepsizes can be used without exact knowledge of $f(x^*)$ in practice, and how sensitive the results are to errors in estimating this value?

---

### Official Review · Reviewer_To1B · 2025-11-10

**Soundness:** 2
**Presentation:** 3
**Contribution:** 2
**Rating:** 4
**Confidence:** 3

**Summary:**

This paper extends the EF21-P and MARINA-P methods to the federated convex nonsmooth problems. For both methods, the authors explore three step sizes: constant, Polyak-type, and decreasing step sizes. They provide theoretical analysis and empirical evidence for both algorithms.

**Strengths:**

* The authors present the first theoretical analysis of the federated convex nonsmooth optimization problem with server-to-worker compression.

* The paper is well-written, and the presentation is clear.

**Weaknesses:**

* I feel that the technical novelty of this paper is limited. Both federated smooth optimization and convex nonsmooth optimization are well-studied areas, and I do not see what specific technical challenges arise in the case of federated convex nonsmooth optimization.

* The constant and Polyak-type stepsize are based on the $x^*$, which is the optimal value of the objective function.

* The numerical experiments are weak. In particular, the empirical experiments are conducted on synthetic nonsmooth functions. How about machine learning experiments with real-world datasets?

**Questions:**

* In related work, the authors claim that recent work with decreasing stepsize can achieve a convergence rate without the log factors on convex nonsmooth problems (lines 123-124). Can the author extend these techniques to the federated setting to achieve an optimal rate without log factors? Is the log factor due to the use of the last iterate instead of the average iterate?

---

### Official Review · Reviewer_da6R · 2025-11-10

**Soundness:** 2
**Presentation:** 1
**Contribution:** 2
**Rating:** 2
**Confidence:** 3

**Summary:**

This paper studies the problem of the optimization problem for nonsmooth federated learning.
They build on existing algorithms EF21-P and MARINA-P for nonsmooth settings.
For EF21-P, they extend it from single-node to a distributed version. Whereas for MARINA-P, they use subgradients to generalize it to nonsmooth optimization.
Both algorithms achieve optimal $O(1/\sqrt{T})$ rate. They also study rates under different step size schemes (constant, decreasing and Polyak).

**Strengths:**

1. Both improved algorithms work in distributed nonsmooth settings, achieving optimal rates (with the caveat that it requires either knowing the total number of iterations 𝑇 in advance for constant stepsize, or knowing the optimal value 𝑓(𝑥*) for Polyak stepsize),
For EP21-P, the communication complexity matches classical distributed subgradient methods.
2. Three different stepsize schemes are studied.

**Weaknesses:**

1. Lack of experiments section in the main text: while in both abstract and contribution section, it is said that experiments show MARINA-P has superior performance, there is no experiment section in the main text. There's an experiment section in the appendix; only synthetic dataset is considered and I do not see comparison with other distributed FL optimization algorithms.
2. Presentation issues: since the author states that MARINA-P has superior performance, the extension of EF21-P seems to be a minor contribution. Whereas for MARINA-P, the only modification the authors made to the algorithm is to use subgradients instead of gradients.
3. Incremental contribution: both algorithms heavily build on previous works, the extensions and the analysis seem relatively standard. It is hard to see whether there are technical contributions in the analysis (related to the presentation issues above).

**Questions:**

1. Is the experiment section supposed to be in the main text, and do you have a comparison with other Fed ML algorithms?
2. What are the technical difficulties when generalizing those two algorithms? Can you summarize the technical contributions in your analysis if there are any?
3. For communication complexity of MARINA-P, you wrote in page 9.
> A notable feature of our complexity result is its independence from
the number of workers 𝑛 in the non-smooth setting – a known phenomenon in subgradient methods
I have trouble understanding why this is a "feature". Does this suggest parallelization provides no benefits, similar what you mentioned in page 7
4. In Algo 1, is line 12 a duplicate?

---

### Note · Authors · 2025-11-23

I have read and agree with the venue's withdrawal policy on behalf of myself and my co-authors.